# On the Mechanical Behaviour of Biosourced Cellular Polymer Manufactured Using Fused Deposition Modelling

**DOI:** 10.3390/polym12112651

**Published:** 2020-11-11

**Authors:** Sofiane Guessasma, Sofiane Belhabib, David Bassir, Hedi Nouri, Samuel Gomes

**Affiliations:** 1INRAE, UR1268 Biopolymères Interactions Assemblages, F-44300 Nantes, France; 2IUMR CNRS GEPEA, Université de Nantes, Oniris, CNRS, GEPEA, UMR 6144, F-44000 Nantes, France; sofiane.belhabib@univ-nantes.fr; 3Borelli Center, ENS Cachan, CNRS, Université Paris-Saclay, 94235 Cachan, France; 4Department of Mechanical Engineering, University Bourgogne Franche Comté (UBFC-UTBM), CNRS/UMR 5060, 90010 Belfort, France; 5Department of traffic engineering, Henan Polytechnical University, No.2001 Shijidadao, Jiaozuo 454003, China; 6IMT Lille-Douai, 941 rue Charles Bourseul, CS 10838, 59508 Douai, France; hedi.nouri@imt-lille-douai.fr; 7LASEM-ENIS, Universite De Sfax, Sfax 3000, Tunisia; 8ICB Laboratory, UMR CNRS 6303, Department COMM, University Bourgogne Franche Comté (UBFC-UTBM), 90010 Belfort, France; Samuel.gomes@utbm.fr

**Keywords:** fused deposition modelling, X-ray micro-tomography, Polylactic acid, cellular structure, finite element computation, compression performance

## Abstract

The aim of this study is to investigate on the compression performance of cellular Polylactic Acid (PLA) manufacturing while using Fused Deposition Modelling. Computer Aided Design (CAD) models of cellular structures are designed using the sequential addition of spherical voids with porosity content varying from 10% to 60%. The three-dimensional (3D) microstructures of cellular PLA are characterised using X-ray micro-tomography to retrieve the correlation between the process-induced defects and the cellular geometrical properties. Mechanical testing is performed under severe compression conditions allowing for the reduction in sample height up to 80%. Finite element computation that is based on real microstructures is used in order to evaluate the effect of defects on the compression performance. The results show a significant drop of the process-induced defects thanks to the use of small layer thickness. Both mechanical anisotropy and performance loss are reduced due to vanishing process-induced defects more significantly when the amount of intended porosities is large. The compression behaviour of 3D printed PLA cellular structures is then found to be only guided by the amount and distribution of the intended porosity.

## 1. Introduction

Additive manufacturing (AM) represents a group of recent processes for the manufacturing of materials using three-dimensional (3D) digitalised models [1,2,3]. According to the review work by Ngo et al. [4], at least six manufacturing processes can be stated as belonging to the class of AM technology [4]. This technology has gained a steepest interest because of the complexity of the forms that can be handled, and the large creativity allowed by the processes without substantial modification of the tooling. Zhai et al., (2014) [5] show that the spread of AM technology is concomitant with the emerging of several AM companies, which took the benefit of the development of several technologies (laser technology, photosensitive solidification, ...), especially during the last three decades [5]. Because AM only requires a limited number of process steps, this allows for it to be more attractive when compared to classical manufacturing techniques, such as moulding regarding fabrication cycles. However, AM processes require the control of numerous process parameters and a higher degree of interactivity with material selection [6]. In fact, the AM processes can be divided into different categories, depending on the initial status of the feedstock material: powder-based such as SLS (Selective Laser Sintering), filament-based (Fused Deposition Modelling, FDM) and liquid-based (stereolithography). Gibson et al. [7] show that the photopolymers developed in the late sixties gave rise to the stereolithography breakthrough in the early eighties thanks to the exposure of UV curable materials to scanning laser [7]. The same authors relate the development of SLS to the extension of plastic-based powder bed fusion to other materials, such as ceramics and metals. Gibson et al. [7] also indicate that extrusion process of polymers that combine pressure and temperature increase was the main route for the development of FDM in the early nineties. Despite the diversity of AM processes, these latter share the same principle of deposition layer-by-layer. In an early contribution, Yan and Gu (1996) [8] show that this is a distinctive feature of AM technologies compared to subtractive techniques where the physical object is formed by 2D layer adding from sliced numerical model [8]. Because of this particular principle of material lying down, some limitations occur due to the way the material is processed. Most of the AM technologies generate material discontinuities that affect the printed part geometry and, thus, its mechanical performance. These discontinuities are intimately related to the slicing step in which filaments are only continuous along their longitudinal length [9]. Choren et al. [10] show that the pore structure has a great influence on the elasticity of printed parts [10]. Guessasma et al. [11] demonstrate that the particular arrangement of the porous structure greatly influences the material behaviour beyond the elasticity stage [11]. In another study, Guessasma et al. [12] show that one of the stress concentration that is induced by the porosity arrangement lowers the mechanical performance, not only under tension, but also under compression [12]. A more recent contribution by Abouzaid et al. [13] report that inhomogeneous deformation of filaments and subsequent cracking patterns are also related to the nature of the porosity network [13]. Indeed, material discontinuities are the sources for the process-induced defects, such as porosities mentioned by several authors. According to Choren et al. [10], these porosities can lead to different types of correlations with the property level, such as Young’s modulus ranging from linear to exponential forms [10]. In addition, Lee et al. [14] report that the level of the compressive strength is significantly dependent on the printing orientation [14]. A recent work by Guessasma et al. [12] shows that the building orientation is correlated to different types of porosity arrangement [11]. Slotwinski and Garboczi [15] report that the methods that are available to measure the porosity level in 3D printed parts, such as mass/volume, CT-scans, Archimedes method, and ultrasounds, all provide good estimation of the porosity content, despite differences in sensitivity and range of detection [15].

The presence of porosities affects both the finishing quality (roughness), volume, and other geometrical attributes in a way that differences may appear between the real part and the CAD (Computer Aided Design) model. Nelaturi et al. [16] conducted a study to measure the deviation between real parts and sliced models. They report that resolution problems appear during the partitioning, leading to significant geometry deviation [16]. In a review paper, Turner and Gold [17] show that several limitations, such as limited motion accuracy, process resolution, variability in feed rate, chordal accuracy, shrinkage, and thermal wrapping, are main factors in determining the dimensional accuracy of a print [17]. In the case of FDM, material discontinuities are generated within the plane of deposition as a difficulty to join properly the fused filaments. Guessasma et al. [11] report CT-scans of polymeric structures that present an evident lack of cohesiveness between filaments [11]. A limited contact between layers is also observed in the building direction as a result of the necking between the filaments. Because the laid down filaments are cylindrical in shape, a layer height equivalent to the diameter of filament generates a single contact point leaving a large space between the layers. This space is part of the defects that were observed in 3D printed parts. Because these defects have a strong outcome on the mechanical anisotropy as well as the loss of performance, it is important to find strategies for decreasing their significance. For instance, the decrease of the layer height to approximately half of the filament diameter, typically 0.2 mm, the contact between the adjacent filaments increases and the inter-filament space becomes smaller. As shown in an early work, Ahn et al. [18] demonstrate that tensile strength is significantly lowered when the filaments are aligned normally to the loading direction because of the material discontinuity between layers [18]. Lee et al. [19] also report lower compressive strength levels in transverse specimens when compared to axial specimens and conclude on anisotropic behaviour [19]. Shaffer et al. [20] confirm the same anisotropic behaviour for tensile specimens and demonstrate the benefit of using ionizing radiation in order to improve the quality of the bonds between filaments [20]. In this work, we propose studying the combined effect of process-induced and CAD-based porosities on the performance of bio-sourced cellular materials manufactured by FDM. The objective is to determine the interaction between the two types of porosities and the consequence on the compression response. The idea of this work comes from the shortcoming that was observed in addressing the link between the microstructure of 3D printed parts and related performance. According to review paper by Gao et al. [6], the focus on process optimisation is considered to be one of the challenges to improve the reliability and cost effectiveness of AM technologies [3]. This challenge was tackled by several researchers and caused a major interest on finding optimal process parameters that fit particular processes and material selection [21,22,23,24]. However, more recent contributions expressed a firm grasp on properly understanding the mechanisms behind the performance such as fatigue [25]. This understanding can be brought by the combination of new 3D imaging techniques, numerical modelling, and validated experimental campaigns as the one proposed in this work.

## 2. Materials and Methods

The CAD models represent cellular materials that are created by adding sequentially macroscopic spherical voids of 8 mm in diameter. The voids are created in a 3D cubic domain of 30 mm in size. Spherical voids are allowed to overlap, hence permitting a large porosity content to be generated. The number of spherical voids controls the final porosity content. The algorithm starts by randomly positioning one spherical void. An attempt to find a position of the second void is made according to density and overlap criteria, as shown in an early paper [26]. The process continues as long as the targeted density is not reached, otherwise, the final design is saved for further processing (i.e., slicing).

Six levels of porosities are obtained from 10% to 60% by a step of 10%. The CAD models are converted into a series of machine instructions using the Ultimaker Cura open source slicing software (Ultimaker, Utrecht, Germany) for further processing. The feedstock material used for 3D printing is the biodegradable plastic PLA (Polylactic Acid) purchased in the form of wires of 2.85 mm in diameter. This material has a density of 1.24 g/cm^3^. The recommended printing conditions by the supplier are: printing temperature (170–220 °C), printing speed (30–80 mm/s), and base temperature (20–60 °C). The FDM equipment Ultimaker Extended 2+ (Ultimaker, Utrecht, Germany) is used for 3D printing of PLA-based cellular structures. The printing temperature is fixed to 220 °C with 100% infill. Soluble support is used in order to maintain the structural integrity of the cellular structures during printing. The orientation of all cellular structures is fixed to 0°, which means that filament crossing forms a sequence of −45°/+45°. A nozzle head of 0.4 mm is used for printing without heating the base, and the vertical resolution (layer thickness) is adjusted to 0.1 mm. The printing rate is 2 mm^3^/s. The cooling rate is kept constant for all realisations according to standard settings (fan power rate 100%). Under these cooling conditions, no material shrinkage or bending of the specimen at the corners are observed. The first built layers for all specimens are sticky to the building platform and no adhesion problems are noticed for all of the printed specimens.

The 3D microstructures of PLA-based cellular structures are imaged while using X-ray micro-tomography (UltraTom X-ray micro-CT from RX Solutions, Chavanod, France). The acquisition parameters are as follows: X-ray source of 230 kV + Al filter, spot size = 24 µm, voltage = 80 KV, current intensity = 300 µA, number of averaging frames = 6, number of radiographic images = 1440, detector resolution = 1920 × 1536 pixels, sample to source distance = 141 mm, sample to detector distance = 579 mm, and voxel size = 31.38 µm. The acquisition duration per sample is 19 min. The tomograms (stack of 2D cross-sections) are built while using a back projection reconstruction algorithm from X-Act software (RX Solutions, Chavanod, France). The typical resolution of the tomograms varies from 1001 × 1002 × 993 voxels to 1034 × 1019 × 1024 voxels, corresponding to physical dimensions of 31.41 × 31.45 × 31.16 mm^3^ to 32.45 × 31.98 × 32.14 mm^3^. Thus, the acquisition volume wraps the entire samples of 30 mm in size. Image analysis is performed on each tomogram in order to derive measurements regarding the porous network using ImageJ software from NIH (Bethesda, MD, USA).

Compression testing is performed while using a universal machine from MTS Systems, Créteil, France (100 KN tensile/compression model). Three replicates per density conditions are considered for mechanical testing. It has to be mentioned that former contributions by authors highlight the high repeatability of the testing results because of the local control of the structure [11]. The compression is performed on all main directions, including Z (building) and lateral (X, Y) directions. The crosshead displacement rate is fixed to 10 mm/min. In order to cope with severe compression conditions, the stages are adapted in order to guarantee full contact with the deformed specimen at any load level. The final compression level is limited by the load cell capacity and it depends on the density of the tested cellular materials. The limit used for the control of the final load is force-based according to a safe limit of 90% of the total load capacity. The force–displacement curves are derived and converted to engineering stress–engineering strain homologue curves. The following engineering quantities are derived: Young’s modulus, elongation at break, yield stress compression strength, and dissipated energy. The dissipated energy is measured from the area under the curve up to the elongation at break limit. Optical recording of the deformed specimens is conducted while using a CMOS camera (Phantom V7.3 from Photon-line, Marly Le Roi, France). The video recording is performed at full resolution (800 × 600 pixels) under a frame rate of 50 fps (frames per second). This setup allows a pixel size of 97 µm.

## 3. Modelling Technique

Finite element modelling is considered to provide a quantification of the defect role on the compression performance of 3D printed PLA cellular structures. This modelling takes the generated defects during the FDM processing into account by implementing the real microstructures of the studied cellular materials. A comparison between the numerical predictions and the experimental results is performed in order to check the consistency of the defect role on the mechanical performance. The conversion of 3D microstructures into meshes uses a principle that is based on voxel-to-element conversion. The meshes are regular and formed by a collection of cubic elements associated to the solid phase. Due to the large resolution of the tomograms and the limited computation resources, a resolution lowering by a factor varying from 5 to 20 is needed to process all of the cellular structures. Under such a condition, the model size varies between 0.15 × 10^6^ and 17.84 × 10^6^ DOF (degrees of freedom), with a typical element size between 157 µm and 628 µm. Translations in the space directions UX, UY, and UZ are the degrees of freedom associated to each node in the mesh.

Uniaxial compression loading conditions are simulated in the three main directions X, Y, and Z. The nodes of the bottom layer are constrained against displacement (UX = UY = UZ = 0). The homologue nodes at the opposite face are displaced by a fixed amount in the loading direction. If the loading direction is X, then (UX = U < 0, UY = 0, UZ = 0). Similar conditions are used for the other loading directions. These loading directions are consistent with the experimental loading, which is conducted without lubrication. This means that the additional constraints on the node are likely to represent the result of the friction contact between the cellular material and compression stages.

The nodal reaction forces in the direction of loading are summed and the effective Young’s modulus is derived for all conditions by solving the linear elasticity problem while using a Preconditioned Conjugate Gradient (PCG). In addition, stress and strain field are predicted for all printed cellular structures. Computations are performed on a workstation that was equipped with 2 CPU operated at 2.4 GHz and 192 Gbytes of RAM. The computation costs are within the range (2–120 min.).

## 4. Results and Discussion

### 4.1. Microstructure and Mechanical Results

Figure 1a depicts the typical compression behaviour of full dense PLA cubes that were manufactured using FDM in all loading directions.

The in-plane deformation sequences in the X and Y directions well demonstrate the resistance of the printed PLA against damage development within the plane of construction, despite the severe load levels. A former study on ABS conducted by Guessasma et al. [11] showed that Poisson’s expansion leads to the emergence of shear bands within plane of construction, which attests that damage growth is possible, even under compression loading [11]. The main deformation mechanism associates the positive strain levels at lateral faces to the process-induced porosities, which are found to connect by pore opening. The present case refers exactly to the same printing conditions (printing angle of 0° promoting filament crossing in a sequence −45°/45°). The deformation sequence for a loading in Z-direction (i.e., building direction) shows tendencies to filament decohesion more evident in this case as compared to the in-plane response. To find out more about the compression behaviour in the case of intended porosity, the deformation sequences for cellular structures containing from 10% up to 60% of intended porosities are shown in Figure 1b.

Despite this similar configuration, the pore opening is inconclusive, even at large load levels. For a loading in the building direction (Figure 1b), there is evidence of ductile behaviour of the printed cellular structures with typical three stages of deformation. The main ones are the cell collapse and densification. There is no significant influence of the raster identified from the deformation sequences. The only minor feature that was observed from the examination of deformed structures in the presence of transverse cracks that develop by a mechanism of crack opening within the plane of construction (see, for instance, the sequences for cellular structures with 40% and 50% of intended porosities). The deformation sequences in the lateral directions (Figure 2 shows examples of loading in Y-direction) clearly depict the presence of vertical cracks.

These correspond to inter-layer delamination involving the building direction. Indeed, the presence of intended porosities generates a stress concentration around the pore and, because of the relatively weak cohesion in the building directions, cracks are likely to depart from the porosity and runs in the vertical direction (normal to the building direction) by a mechanism of crack opening. This mechanism is possible because of the Poisson’s expansion.

In terms of load response, Figure 2 compares the stress-strain curves that correspond to the deformation sequences in Figure 1. Stress and strain values are expressed as engineering quantities.

The three stages of deformation of a ductile cellular material are depicted, even for the case of a fully dense PLA structure (Figure 2a). The mechanical response is not similar for all deformation stages with the exception of the elastic stage. Indeed, the slope seems to be similar for all loading directions. The Young’s modulus obtained for the full dense PLA suggests that there is no mechanical anisotropy that is associated with the printing of PLA under the conducted printing conditions (Table 1). The reason behind the similarity in elasticity behaviour relies on the amount of the porosity. This amount, if measured in all space directions, would show the same score. This would explain the minor difference of 8% in Young’s moduli measured in X, Y, and Z directions. A more elaborated discussion of porosity content is suggested in the next section.

The difference between Young’s moduli is less than 8%. For a further increase of the load level, the 3D printed PLA exhibits a superior trend in the building direction (Z-direction). The development of damage that is evidenced from the deformation sequence (Figure 1a) corresponds to a decrease in the engineering stress for the engineering strain between 0.28 and 0.4. From the reading of yield stress and compression strength magnitudes in Table 1, it can be stated that the good performance of the 3D printed PLA loaded in the building direction is obtained.

An improvement by 30% and 21% of these quantities is achieved with respect to the in-plane properties. This improvement is explained, from the processing viewpoint, by the small layer thickness used for building the printed PLA, which only represents 25% of the nozzle diameter.

Figure 2b,c compare the compression response of the printed PLA cellular structures for all intended porosity contents. In the loading direction, the trend ranking goes with the porosity amount. Low rankings in terms of elasticity, yielding, and strength are evidenced for large porosity contents. A positive effect of the intended porosity content on the width of the collapse plateau compensates the loss of performance with the increase of intended porosity content. The damage development within the cell collapse plateau depicted through the deformation sequences in Figure 1b seems to have a limited effect on the cell collapse plateau. This effect identified as transverse cracking is to be related to the flatness of this deformation stage. However, the loading in the lateral direction Y (Figure 2c) shows different compression responses. Indeed, within the second deformation stage, cell collapse is associated with significant decrease in the engineering stress, which is contrasted with the same deformation stage for a loading in the building direction (Figure 2b). The development of vertical cracks due to the inter-layer delamination has a more significant effect on the compression response. This effect seems to be stronger when the intended porosity content is low, such as for 10%.

The quantification of the intended porosity content of 3D printed PLA cellular structures on extracted mechanical parameters, namely Young’s modulus, yield stress, compression strength, and mechanical energy, are depicted in Figure 3 for all loading directions.

If we look at the first three mechanical parameters, there is no mechanical anisotropy identified from the loading in all directions if we exclude Young’s moduli measured at 10% of intended porosity. The fitting performed for all data shows the following linear trends
(1)$$E\left(\mathrm{GPa}\right)\;=\;1.39-0.02\times f\left(\%\right);R^{2}\;=\;0.97$$
(2)$$\sigma_{Y}\left(\mathrm{MPa}\right)\;=\;52-0.85\times f\left(\%\right);R^{2}\;=\;0.93$$
(3)$$\sigma_{C}\left(\mathrm{MPa}\right)\;=\;60-0.97\times f\left(\%\right);R^{2}\;=\;0.93$$
where E is Young’s modulus, $\sigma_{Y}$ is yield stress, $\sigma_{C}$ is the compression strength, and f is the intended porosity content varied between 10% and 60%.

The correlation between the mechanical parameters of 3D printed cellular PLA and the intended porosity content suggests simple linear forms when compared to the exponential functions generally associated with the behaviour of typical cellular materials [27]. In addition, the extrapolation of the mechanical properties for a full dense 3D printed PLA overestimates the magnitudes that were reported in Table 2. This is due to the fact that process-induced porosity is neglected in expressions (1) to (3).

Figure 3d shows that the amount of dissipated energy during the compression loading is larger for a loading in Z-direction. This is true, especially when the amount of intended porosity is low.

The mechanical anisotropy can be considered to be low, even for the energy dissipation with regards to the relative small differences between loading directions.

The overall trend of energy dissipation, irrespective of the loading direction, is linear as in the previous cases
(4)$$\varphi \;\left(\mathrm{mJ}/{\mathrm{mm}}^{3}\right)\;=\;39-0.59\times f\left(\%\right);R^{2}\;=\;0.98$$
where φ is the amount of dissipated mechanical energy that us associated with a compression by a 60% reduction in height.

In order to provide a microstructural interpretation of the mechanical results, X-ray micro-tomography images are exploited in the following. Figure 4a shows cross-section and perspective views of the porosity that is induced by processing for a dense 3D printed PLA structure. A regular grid of micro-sized porosities is evidenced. From the cross-section views, the pore connectivity within the plane of construction (XY) follows the raster orientation by a sequence of −45°/+45°. Additionally, small porosities are identified between the raster and the external frame. These emerge as a consequence of the abrupt change in the printing nozzle trajectory during the laying down process. In the orthogonal planes XZ and YZ, pore connectivity in the building direction is also a main characteristic of the process-generated defects. Table 1 shows the statistics conducted on the pore amount and connectivity. These data demonstrate the main characteristic feature of 3D printed materials using FDM, which is the formation of a highly connected regular network of micro-sized porosities, despite their small amount (Figure 4b).

Previous data on ABS show that the pore connectivity, in this case, is small when compared to the massively connected pore networks of 85% reached for ABS [28]. This difference is mainly due to the smaller ratio between the layer thickness and the nozzle diameter used in this study, which is four times in proportion lower than the one that was used for ABS.

Figure 4c displays the content profiles of the porosity induced by processing in the full dense 3D printed PLA. These profiles in the main three directions show two main features. The first one is the significant oscillation of the porosity content that indicates the regularity of pore spatial distribution in all directions. The second feature is the presence of characteristic peaks at endpoints in X-direction, which indicate the presence of porosity between the frame and the raster reaching more than 8%. The same peaks should be observed in Y-directions if the cropping wraps the entire sample width.

Figure 5a,b depict the cross-section views normal to X, Y, and Z directions for all PLA 3D printed cellular structures. These cross-section views show the relative absence of process-induced porosity, especially when the amount of intended porosity is large. For instance, minor micro-sized porosities that follow the raster orientation are found within the bulk for PLA900 and PLA800 corresponding to 10% and 20% of intended porosities (Figure 5a). Additionally, porosities between the raster and the frame are also visible, especially for PLA900. In addition, defects near the intended porosities are also depicted.

These are created because of the strategy used to build the contour of the porosities themselves. These contours leave some space between the raster and the porosity frontiers. The process-induced porosities within the raster substantially decrease for PLA690, referring to 30% of intended porosities. However, some defects that are close to the external frame and at the intended porosities boundaries are still persistent. When a larger content of intended porosities is created in the 3D printed PLA matrix (Figure 5b), the amount of defects vanishes, including the process-induced porosities and the defects at the boundaries.

Figure 5c illustrates the extent of defect through perspective views of all 3D printed PLA cellular structures. These cellular materials exhibit a cohesive structure and the laying generally featured in the building direction is absent. This is an additional proof of the beneficial role of a low layer thickness.

Figure 6 highlights the difference between the porosity profiles in order to evaluate the rendering of the 3D printed cellular structures with regards to the expectations from the CAD models. These porosity content profiles are plotted in the building directions. Both cellular structures with low (Figure 6a) and large (Figure 6b) contents of intended porosities depict the same minor difference between CAD models and 3D printed features.

This means that, under the printing conditions used in this study, an acceptable accuracy for printability of cellular PLA is obtained. Table 1 shows that the overall porosity content of the cellular structures does not vary significantly from the intended one. The difference between the intended and measured porosity contents varies between 5% and 17%. The lowest differences correspond to the cellular materials with the largest amount of porosity contents. The connectivity of the pores increases with the increase of intended porosity contents reaching 99% for PLA390. At the same time, this connectivity proves to be substantially reduced for small intended porosity content, reaching low levels of 22% for PLA900. This level contrasts with the 65% that was obtained for ABS for the same amount of intended porosity [28].

### 4.2. Finite Element Results

Finite element computation is used in order to predict the overall behaviour of 3D printed PLA cellular structures and find out more about the deformation mechanisms in the presence of process-generated defects. In fact, the combination of the two porosities prove to be the key to understand how stress concentrators contribute to the overall performance of the printed materials and more specifically to the failure modes. When the connectivity between the generated and process-induced porosities is varied, the cracking initiation and propagation may differ. With higher connectivity, instable cracking is likely to occur, while a more discontinuous porous structure triggers more diffuse cracking. This is, for instance, observed for fully dense structures that were printed under different orientations [11].

Figure 7a shows the predicted stress intensity counterplots for a fully dense PLA structure that was loaded in different directions and under different resolutions. The load level corresponds to 1% in height reduction. The change in resolution is performed by varying the element size in the regular meshing through the voxel size. A low resolution is associated with a large voxel size. Because of the computation resource limitation, the original resolution (voxel size = 31.4 µm) could not be handled. Indeed, such a voxel size leads to the implementation of models with more than one billion of degrees of freedom. The smallest voxel size that can be reached is 157 µm, which is nearly three times smaller than the printing resolution (i.e., nozzle diameter). This value represents the best compromise that allows for details of the 3D microstructural arrangement of process-induced porosity to be preserved. It can be seen from Figure 7b that the stress heterogeneity is guided by the regularity of the spatial distribution of process-induced porosity. This latter acts as stress concentrators with an extent that depends on the level accuracy determined by the voxel size. For a large voxel size, such as 628 µm, the regions of large stress intensity are discontinuous, and one can barely retrieve the effect of raster orientation on the stress localisation. More continuous stress heterogeneity is predicted with larger voxel sizes allowing for the smallest one to capture the main characteristics of the stress variation within the raster and across the building direction.

In addition to such stress heterogeneity, the regions of highest stresses are also predicted at the edges according to the loading conditions. Figure 7b,c show the same stress intensity counterplots for all 3D printed PLA cellular structures for all loading situations that were conducted under the largest possible resolution (voxel size = 157 µm). These counterplots show no significant effect of the process-induced porosity as in the former case even for cellular structures with a small amount of intended porosities (Figure 7b). The major stress heterogeneity is determined by the presence of intended porosities, which are the major stress concentrators in the cellular structures. The variability in stress magnitude is more pronounced for the case of larger intended porosity content (Figure 7c). The predicted Young’s modulus that was derived from these computations plotted as a function of the resolution in Figure 8a. A remarkable stability of this predicted property is noticeable across the studied resolutions, despite the change in microstructural details that are evidenced in Figure 7b,c.

Additionally, no major differences between the loading directions are observed. Most of the changes are related to the fully dense PLA, which is found to be more sensitive to the resolution lowering when compared to the cellular materials.

The comparison between the experimental and numerical Young’s moduli indicates that there is a fair agreement between the two results, despite the scatter depicted in Figure 7c. It can be concluded that finite element computations relying on continuity between filaments are sufficient hypotheses to lead to realistic predictions. This also means that there is no need for the implementation of weak links at the junction between filaments, because of the relative cohesive structure that was induced by the small layer thickness.

## 5. Conclusions

This study concludes that the 3D printing of PLA using a layer thickness as small as a fourth of the nozzle diameter leads to a substantial reduction of the effect of process-induced porosity. The full dense PLA printed under these conditions generate approximately 5% of porosity, which tends to be of low connectivity (65%) when compared to reported studies on ABS. The effect of these defects is substantially reduced when cellular structures are manufactured with a low porosity amount of less than 20%. For higher intended porosity rates, the effect of process-induced porosity completely vanishes. From the mechanical performance viewpoint, this has a direct consequence on the reduction of mechanical anisotropy and the low extent of damage induced by defects. The results achieved in this study are promising with regards to the free of defect manufacturing promoted by the use of small layer thickness. This unusual material continuity obtained for FDM is only to be compared with other types of additive manufacturing that are known to generate no material discontinuity, such as stereolithography. The main advantage when compared to stereolithography is the large spectrum of materials that can be processed using FDM. The only drawback of decreasing the layer thickness remains the process duration that goes with the requirement of achieving a relatively free of defect parts.

## Figures and Tables

**Figure 1 polymers-12-02651-f001:**
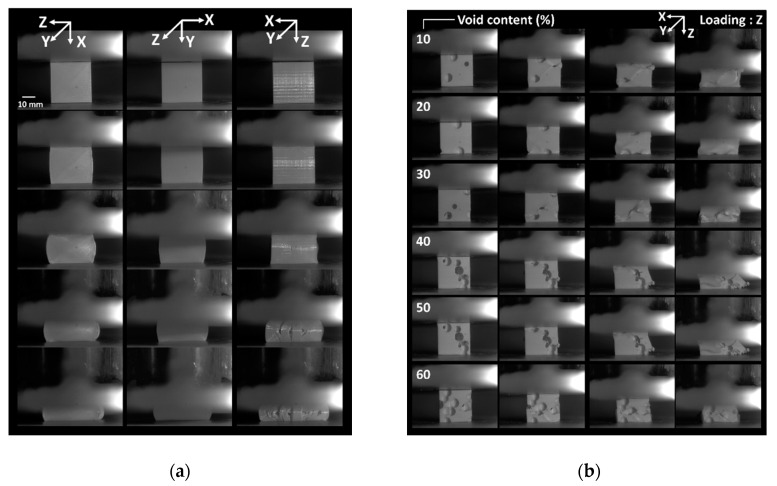
Compression results of airy printed Polylactic Acid (PLA) structures: (**a**) compression sequence of dense PLA printed structures in different loading directions. Z is the building direction, (**b**) deformation sequences as a function of the porosity content of three-dimensional (3D) printed cellular PLA. Loading is performed in Z-direction.

**Figure 2 polymers-12-02651-f002:**
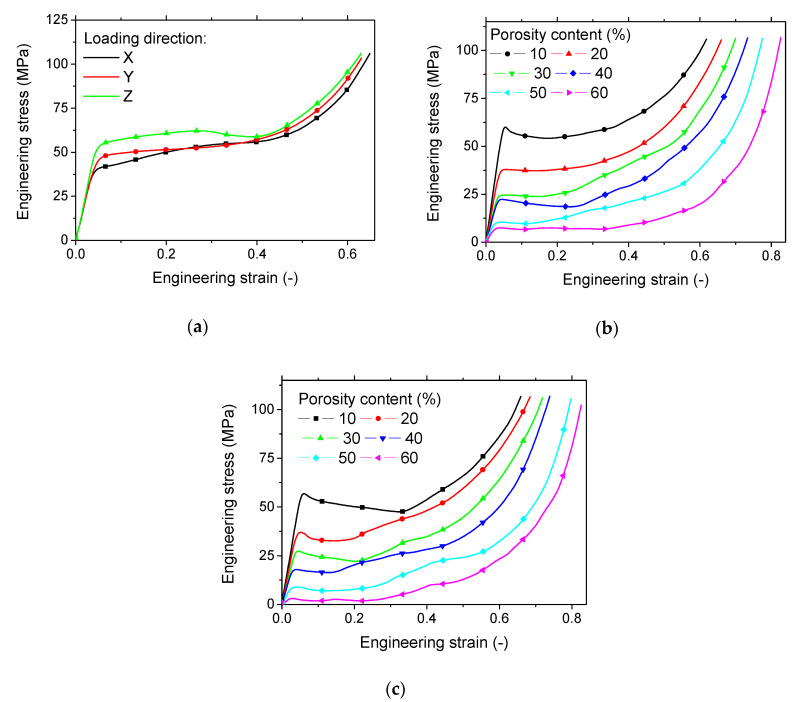
Compression response of (**a**) 3D printed dense PLA in all loading directions, and 3D printed cellular PLA as a function of the porosity content for loading in (**b**) Z- and (**c**) Y-directions. Z is the building direction.

**Figure 3 polymers-12-02651-f003:**
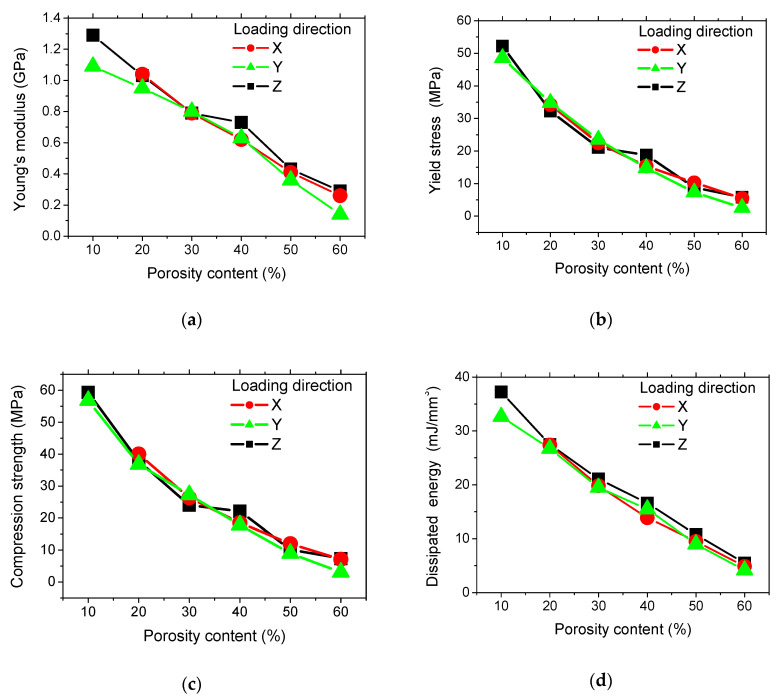
Evolution of mechanical properties as a function of the porosity content for 3D printed cellular PLA. (**a**) Young’s modulus, (**b**) yield stress, (**c**) compression strength, and (**d**) dissipated mechanical energy.

**Figure 4 polymers-12-02651-f004:**
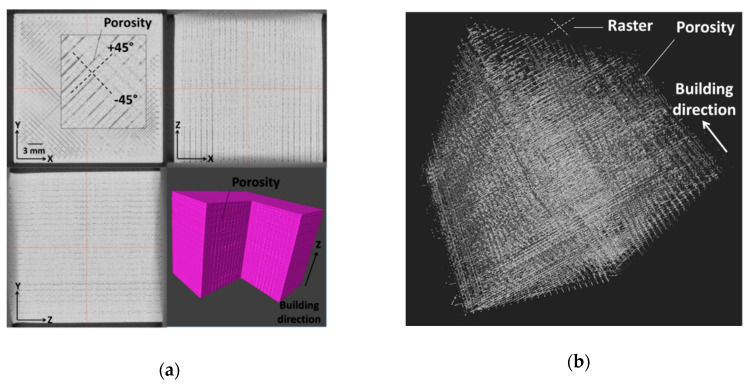
X-ray microtomography results (**a**) Cross-section and perspective views of 3D printed dense PLA imaged using X-ray micro-tomography, (**b**) porous network of 3D printed dense PLA and (**c**) related porosity content profiles in the main directions of the structure.

**Figure 5 polymers-12-02651-f005:**
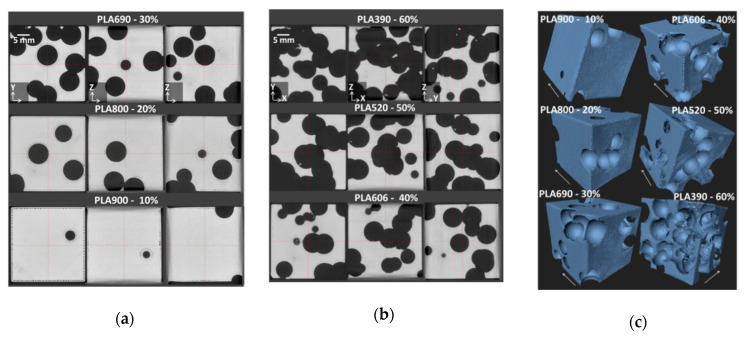
Cross-section views in the planes XY, XZ, and YZ from X-ray micro-tomography 3D images showing the cellular structure of 3D printed cellular PLA. (**a**) low porosity range 10–30%, and (**b**) large porosity range 40–60%, (**c**) Perspective views of 3D printed PLA cellular structures showing process-induced defects. The arrows indicate the building direction.

**Figure 6 polymers-12-02651-f006:**
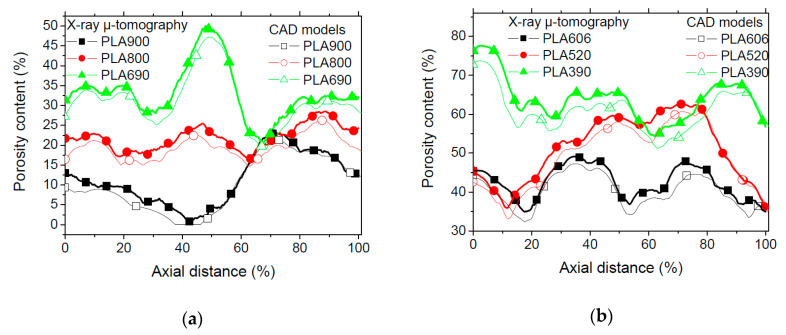
Comparison between porosity content profiles of Computer Aided Design (CAD) and X-ray micro-tomography along the building direction for 3D printed PLA cellular materials exhibiting (**a**) low (<40%) and (**b**) large (>30%) porosity contents.

**Figure 7 polymers-12-02651-f007:**
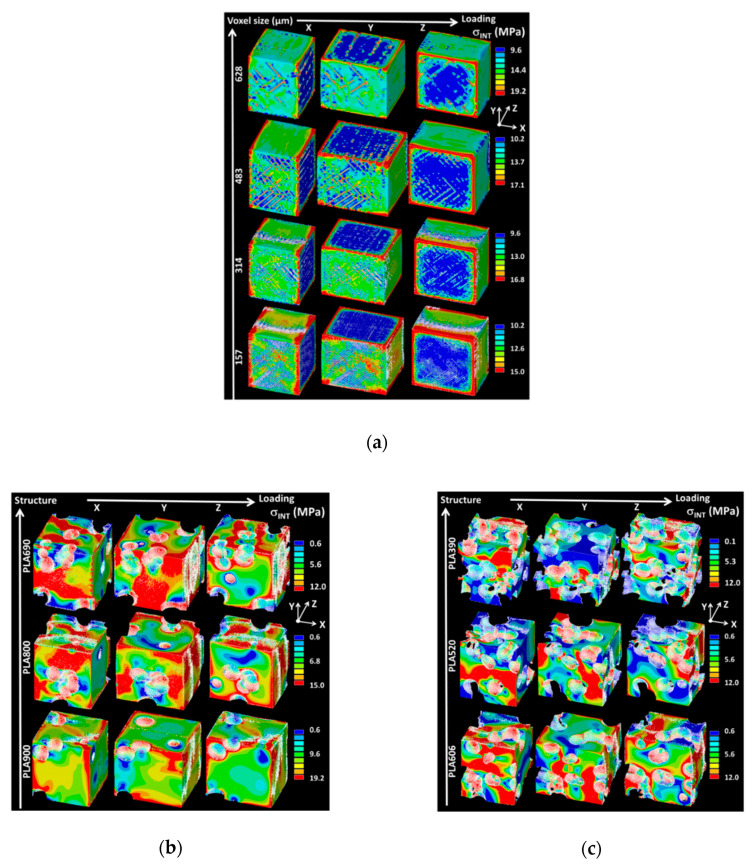
Finite element results (**a**) Stress intensity (σ_INT_) counterplots as a function of loading direction and voxel size for a fully dense configuration of 3D printed PLA. Stress intensity (σ_INT_) counterplots as a function of loading direction of 3D printed PLA cellular structures exhibiting (**b**) low (<40%) and (**c**) large (>30%) porosity contents.

**Figure 8 polymers-12-02651-f008:**
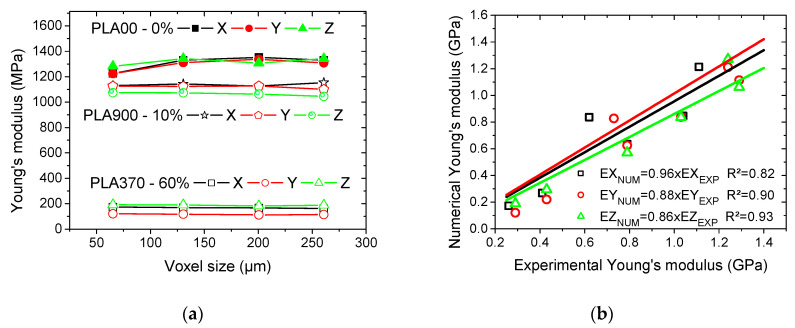
Predicted finite element results: (**a**) predicted Young’s modulus as a function of voxel size in all loading directions for various 3D printed PLA cellular structures, (**b**) comparison between predicted and experimental Young’s modulus for all 3D printed PLA cellular structures.

**Table 1 polymers-12-02651-t001:** Compression properties of fully dense 3D printed PLA.

Sample	Loading	Young’s Modulus (GPa)	Yield Stress (MPa)	Compression Strength (MPa)	Plateau Width (−)	Mechanical Energy (mJ/mm^3^) *
PLA00X	X	1.11	32.33	40.56	0.45	32
PLA00Y	Y	1.07	35.70	48.12	0.43	33
PLA00Z	Z	1.24	45.44	54.41	0.42	37

* up to 60% of reduction in height.

**Table 2 polymers-12-02651-t002:** Summary of main structural parameters associated with 3D printed cellular PLA.

Sample	Intended Porosity Content (%)	Measured Porosity Content (%)	Measured Pore Connectivity (%)
PLA000	0	5.45	64.73
PLA900	10.00	11.74	21.89
PLA800	19.30	21.94	38.33
PLA690	30.90	33.12	22.48
PLA606	39.30	42.46	73.10
PLA520	47.70	51.43	95.83
PLA390	60.70	63.98	99.34

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
