# Peer review of "On the Mechanical Behaviour of Biosourced Cellular Polymer Manufactured Using Fused Deposition Modelling"

_polymers, 2020, doi:10.3390/polym12112651_

Round 1
Reviewer 1 Report
In this paper, the authors investigated the effect of the nozzle diameter on the porosity of 3D printed PLA and succeeded the reduction of process-porosity of full dense PLA under manufacturing with FDM. The results were discussed quantitatively with Young’s modulus, Yield stress, Compression strength, Plateau width and Mechanical energy. The authors collected enough data about properties of PLA to conclude the relationship between the porosity of PLA and the manufacture. So, this paper contains important sight in the 3D printed PLA field and gives the researchers new design guideline. I believe strongly this paper should be published in the journal, Polymers.
Small questions
1. What is the mechanism that thinner layer thickness reduces porosity? Is that simple reason thin layer can fill the space high-densely?
2. Did the authors control the cooling rate? There is a report that the cool rate induces the shrink of material layers.
Author Response
Reviewer 1 :
In this paper, the authors investigated the effect of the nozzle diameter on the porosity of 3D printed PLA and succeeded the reduction of process-porosity of full dense PLA under manufacturing with FDM. The results were discussed quantitatively with Young’s modulus, Yield stress, Compression strength, Plateau width and Mechanical energy. The authors collected enough data about properties of PLA to conclude the relationship between the porosity of PLA and the manufacture. So, this paper contains important sight in the 3D printed PLA field and gives the researchers new design guideline. I believe strongly this paper should be published in the journal, Polymers.
Small questions
- What is the mechanism that thinner layer thickness reduces porosity? Is that simple reason thin layer can fill the space high-densely?
This is an interesting comment from the reviewer. The link between the layer height and the density is related to the necking effect. As the filaments are cylindrical in shape, a layer height equivalent to the diameter of filament generates a single contact point leaving a larger space between the layers. By decreasing the layer height to approximately half of the filament diameter, typically 0.2 mm, the contact between the adjacent filaments increases and the inter-filament space becomes smaller.
A comment was made to provide such an explanation in the new version.
Page 2: “Because the laid down filaments are cylindrical in shape, a layer height equivalent to the diameter of filament generates a single contact point leaving a large space between the layers. This space is part of the defects observed in 3D printed parts. Because these defects have a strong outcome on the mechanical anisotropy as well as the loss of performance, it is important to find strategies to decrease their significance. For instance, the decrease of the layer height to approximately half of the filament diameter, typically 0.2 mm, the contact between the adjacent filaments increases and the inter-filament space becomes smaller.”
- Did the authors control the cooling rate? There is a report that the cool rate induces the shrink of material layers.
We agree with the reviewer, the cooling rate has an effect on the thermal cycling and thus on the residual stresses and material shrinkage. It has a significant effect on the first built layers especially for large realisation. Bending of the specimen at the corner is generally observed for those parts with a large surface contact area with the building platform. In our experiment we kept the cooling rate to its reference value. We did not observe any adhesion problems for all the printed specimens.
Amendment in page 3: “The cooling rate is kept constant for all realisations according to standard settings (fan power rate 100%). Under these cooling conditions, no material shrinkage, nor bending of the specimen at the corners are observed. The first built layers for all specimens are sticky to the building platform and no adhesion problems are noticed for all the printed specimens.”

Reviewer 2 Report
1. First reference was [11]?
2. The author summarized numerous current investigations. What were the shortcomings of these studies? Without this information, it's hard to judge or believe the authors’ work was meaningful. Why was it important to determine the interaction between two types of porosities and the effects on compression response? What’re the significance?
3. How to accurately control the number of spherical voids and porosity during printing?
4. Figure 1 was very confusing. What’re the x, y, and z directions? Please add schematic to show the coordinate system.
5. How many samples were measured in compression tests? Without average and standard deviation, it is hard to say the trends summarized by the authors were correct or not.
6. Why the loading from different directions had the same/similar results in compression testing. The filament orientation should be different in vertical and horizontal directions, which resulted in different mechanical properties when loading along different directions. Any explanations?
7. How to calculate the mechanical energy? This information was missing in Materials and Methods.
8. How to distinguish the curves of PLA900-10% from PLA00-0% in Fig. 8a
Author Response
- First reference was [11]?
We are sorry for the renumbering problem. This is now fixed using automatic numbering.
- The author summarized numerous current investigations. What were the shortcomings of these studies? Without this information, it's hard to judge or believe the authors’ work was meaningful.
We agree with the reviewer to bring to the reader attention the shortcomings of the cited contributions. In this new version, we added the missing link between the microstructure, the process parameters and the mechanical performance. One can read in page 2:
Page 2: “However, AM processes require the control of numerous process parameters and higher degree of interactivity with material selection (Gao et al.2015)”.
Page 3: “The idea of this work comes from the shortcoming observed in addressing the link between the microstructure of 3D printed parts and related performance. According to review paper by Gao et al. [6], the focus on process optimisation is considered as one of the challenges to improve the reliability and cost effectiveness of AM technologies [3]. This challenge was tackled by several researchers and caused a major interest on finding optimal process parameters that fit particular processes and material selection [21-24]. But more recent contributions expressed a firm grasp on properly understanding the mechanisms behind the performance such as fatigue [25]. This understanding can be brought by the combination of new 3D imaging techniques, numerical modelling and validated experimental campaigns as the one proposed in this work.”
Why was it important to determine the interaction between two types of porosities and the effects on compression response? What’re the significance?
This is a very important point tackled by the reviewer. In fact, the combination of the two porosities prove to be the key to understand how stress concentrators contribute to the overall performance of the printed materials and more specifically to the failure modes. When the connectivity between the generated and process-induced porosities is varied, the cracking initiation and propagation differ. With higher connectivity, instable cracking is likely to occur while more discontinuous porous structure trigger more diffuse cracking.
Amendment in page 11: “In fact, the combination of the two porosities prove to be the key to understand how stress concentrators contribute to the overall performance of the printed materials and more specifically to the failure modes. When the connectivity between the generated and process-induced porosities is varied, the cracking initiation and propagation may differ. With higher connectivity, instable cracking is likely to occur while more discontinuous porous structure trigger more diffuse cracking. This is, for instance, observed for fully dense structures printed under different orientations [11].”
- How to accurately control the number of spherical voids and porosity during printing?
The spherical voids are designed according to a sequential addition algorithm as stated in the previous version. The algorithm starts by randomly positioning one spherical void. An attempt to find a position of the second void is made according to density and overlap criteria as shown in an early paper. The process continues as long as the targeted density is not reached, otherwise, the final design is saved for further processing (i.e., slicing). So, the number of spheres is controlled prior printing.
Amendment in page 3: “. The algorithm starts by randomly positioning one spherical void. An attempt to find a position of the second void is made according to density and overlap criteria as shown in an early paper [26]. The process continues as long as the targeted density is not reached, otherwise, the final design is saved for further processing (i.e., slicing).”
- Figure 1 was very confusing. What’re the x, y, and z directions? Please add schematic to show the coordinate system.
The coordinate system is added in both figures 1a and 1b.
- How many samples were measured in compression tests? Without average and standard deviation, it is hard to say the trends summarized by the authors were correct or not.
We repeated sufficiently the experiments to be confident in the data we observed. We performed 3 replicates per density condition. In addition, a particular feature of 3D printed part that we highlighted in several publications is the high repeatability of the testing results because of the local control of the structure. The reviewer can check the following result showing differences less than 2% for samples printed using the same FDM conditions.
Difference between mechanical responses of samples printed using the same process conditions. Porosity content is 10%, same positioning of the spherical voids.
Amendment page 4: “Three replicates per density conditions are considered for mechanical testing. It has to be mentioned that former contributions by authors highlight the high repeatability of the testing results because of the local control of the structure [11].”
- Why the loading from different directions had the same/similar results in compression testing. The filament orientation should be different in vertical and horizontal directions, which resulted in different mechanical properties when loading along different directions. Any explanations?
We did not mention that the mechanical response is similar in all stages, only for the elasticity stage. The reason is that the elasticity behaviour relies on the amount of the porosity. This amount is the same if measured in all directions and this is attested by the 8% of difference in Young’s modulus. The examination of Figure 1 shows some differences in the collapse and densification stages.
Amendment in page 7: “The mechanical response is not similar for all deformation stages with the exception of the elastic stage. Indeed, the slope seems to be similar for all loading directions. Young’s modulus obtained for the full dense PLA suggests that there is no mechanical anisotropy associated with the printing of PLA under the conducted printing conditions (Table 1). The reason behind the similarity in elasticity behaviour relies on the amount of the porosity. This amount if measured in all space directions would show the same score. This would explain the minor difference of 8% in Young’s moduli measured in X, Y and Z directions. A more elaborated discussion of porosity content is suggested in the next section.”
- How to calculate the mechanical energy? This information was missing in Materials and Methods.
Sorry for this omission. Indeed, the energy is measured from the integral under the curve up to the elongation at break.
Amendment in page 4: “The force – displacement curves are derived and converted to engineering stress – engineering strain homologue curves. The following engineering quantities are derived: Young’s modulus, elongation at break, yield stress compression strength, and dissipated energy. The dissipated energy is measured from the area under the curve up to the elongation at break limit.”
- How to distinguish the curves of PLA900-10% from PLA00-0% in Fig. 8a
The reviewer is right. The symbols used are the same. However, we specified the conditions next to the corresponding curves. To avoid the confusion, we modified the symbols.
Please check the updated version of Figure 8.

Round 2
Reviewer 2 Report
I have no comments for this revised manuscript.